

# Effects of different light intensities on lettuce growth, yield, and energy consumption optimization under uniform lighting conditions

Jun Zou[1], Shipeng Luo[1], Mingming Shi[1], Dawei Wang[1], Wenbin Liu[1], Yan Shen[2], Xiaotao Ding[3] and Yuping Jiang[4]

[1] Faculty of Science, Shanghai Institute of Technology, Shanghai, Fengxian, China
[2] Shanghai Yingzhi Technology Co., Ltd, Shanghai, Fengxian, China
[3] Shanghai Academy of Agricultural Sciences, Shanghai, Fengxian, China
[4] College of Ecological Technology and Engineering, Shanghai Institute of Technology, Shanghai, Fengxian, China

## ABSTRACT

Vertical farming is an advanced form of modern agriculture, but it involves high energy consumption when providing supplemental lighting for crops. The research designed an automated lighting detection device and explored its application in vertical farms. It comprehensively investigated the effects of different red-blue light intensities on two lettuces (*Lactuca sativa* L. cv. 'Spanish green' and 'Butterhead') varieties under highly uniform supplemental lighting conditions. The research encompasses investigations into lettuce morphological parameters, photosynthetic physiology, productivity, and energy consumption. The experimental light quality and photoperiod were set at (R/B = 4, 16 h/d), with light intensity ranging from 125 to 300 μmol/m$^2$/s. The experiment indicates that with the increase in light intensity, the yield of both lettuce varieties also increases. Furthermore, lettuce exhibits a significant increase in leaf amplitude from day 20 to day 25 of growth. At 300 μmol/m$^2$/s, Butterhead lettuce demonstrates optimal fresh weight and productivity ratio. As for Spanish green lettuce, the productivity ratio at 200 μmol/m$^2$/s is only 2.11% lower than at 300 μmol/m$^2$/s. Under the cultivation conditions of R/B = 4 and 16 h/d, a light intensity of 300 μmol/m$^2$/s is suitable for the production of Butterhead lettuce, in comparison, a light intensity of 200 μmol/m$^2$/s is suitable for the production of Spanish green lettuce.

## INTRODUCTION

Light is one of the important environmental factors for plant growth, which influences photosynthesis and induces various physiological responses in plants, such as germination, flowering, and fruiting (*Abidi et al., 2013*; *Zushi, Suehara & Shirai, 2020*; *Givens et al., 2023*). The current research broke light down into three components: light quality, light intensity, and photoperiod, which significantly impact plant growth and development

Corresponding authors
Jun Zou, zoujun@sit.edu.cn
Mingming Shi, mmshi@sit.edu.cn

(*Kang et al., 2013*; *Zhang et al., 2018*). Compared with the soil cultivation of traditional agriculture, the current mainstream soilless culture technology promotes the development of modern agriculture. Hydroponics is a commonly used method in soilless culture. It supplies nutrients to plants by adding a nutrient solution and soaking the roots, providing a strong foundation for the development of facility agriculture. Vertical agriculture is an advanced form of modern agriculture, which realizes high yield by increasing the yield of limited space plane (*Engler & Krarti, 2021*; *Delorme & Santini, 2022*). Vertical agriculture is generally applicable in greenhouse environments and is highly dependent on soilless cultivation techniques and environmental control technologies. This means that it requires more electricity consumption, and lighting electricity consumption in vertical agriculture is an important factor affecting its economy (*Weidner, Yang & Hamm, 2021*). In vertical farming, major expenses include depreciation, labor, and electricity costs, with electricity often accounting for 20% to 30% of the total expenses (*Kozai & Japan Plant Factory Association, 2019*). Spectrum adjustment technology can help reduce production costs to some extent, but different plants have different needs for spectra, photosynthetic photon flux density (PPFD), and photoperiods, which results in plant factories are often unable to grow plants in an optimally regulated manner (*Iersel & Gianino, 2017*; *Wang et al., 2016*).

Currently, plant lighting primarily adopts LED lighting. The main types include top lighting, lateral lighting (*Mutombo Arcel et al., 2023*; *Chen et al., 2023*), bottom lighting (*Zhang et al., 2015*; *Joshi et al., 2017*), and inter-plant lighting in enclosed artificial environments. To facilitate the operation of plant factories, top lighting is commonly used. In terms of shape, plant lamps are available in bar, round, and square designs. Among them, bar lamps, typically in the form of long straight tube lamps, are widely adopted in plant factories due to their compact size and easy assembly. Round plant lamps are mainly used in greenhouses, while square plant lamps are suitable for small-scale plant factories, where their concentrated light intensity ensures high uniformity over a small area.

Light intensity has an important effect on photosynthesis in plants. In a wide range of studies, the photosynthetic response curve of plants reaches an equilibrium point with increasing light intensity (*Poorter et al., 2019*), this means that light intensity above that point no longer increases photosynthesis in plants. Within a certain range, the rate of photosynthesis in plants increases with increasing light intensity. Too low light intensity can induce shade avoidance responses in plants, resulting in thin stems and leaves (*Yang et al., 2014*; *Gong et al., 2015*; *Shafiq et al., 2021*). Lettuce, the main leafy vegetable grown in plant factories, has the advantages of a short growth cycle and high yield (*Zhang et al., 2018*; *Bantis et al., 2018*), and adjusting the light can significantly affect the growth, yield and quality of lettuce. Experiments on lettuce plants under seven different light intensities were conducted in the study by *Grzegorzewska et al. (2023)* and showed that lettuce quality was better at a PPFD of 200 µmol/m$^2$/s compared to 160 µmol/m$^2$/s. Many studies have shown the importance of light intensity for lettuce growing, but in the study of optimal light intensity for lettuce, the values of light intensity at which the experimental groups were placed were only roughly given. In terms of light intensity, the main parameter of interest is the PPFD, which is highly influenced by distance. In addition, the uniformity of light also needs to be considered. Variations in the spatial distribution of light uniformity

can lead to significant differences in crop growth (*Boros et al., 2023*), which is crucial for maintaining consistent crop growth both in commercial production and experimental research. While there has been some research on the design and calculation methods for the uniformity of lighting fixtures (*Hwa-Soo, Sook-Youn & Jae-Hyun, 2014*; *Hitz et al., 2019*; *Balázs et al., 2022*), it has not been widely addressed in most supplemental lighting studies for crops. This study provides a measuring tool and luminaire structure, which is used to accurately go through the distribution of PPFD of the luminaire, and then manually adjust the lighting angle of the lamp to ensure that the plane of the lamps is at a high PPFD uniformity, on the basis of which the effects of different light intensities on lettuce growth are investigated and the optimal light intensity suitable for the production of lettuce in vertical agriculture plant factories is determined in relation to the power consumption of the lamps.

## MATERIALS AND METHODS

### Vertical cultivation structure and control environment

The vertical cultivation rack consists of four layers, constructed with aluminum alloy material to form the overall framework, with specific dimensions of [2.6 m (length) × 1.28 m (width) × 1.8 m (height)], each layer is equipped with 12 adjustable LED tubes (T8) for controlling the irradiation angle. The light intensity is regulated through a control panel. Additionally, cross-flow fans are placed on one side of each layer to reduce the heat generated by the lamps. The cross-flow fans operate simultaneously with the lamps when they are turned on. Each layer is fitted with foam boards evenly distributed with planting holes, accommodating a total of 84 lettuce plants. Underneath each cultivation rack, there are water tanks and water pumps. The water pump draws the nutrient solution to the fourth layer, and then the solution flows to the other layers through gravity. When the water level in the tanks drops below the position of the pump motors, nutrient solution is replenished. The temperature throughout the entire laboratory is maintained through constant temperature control *via* air conditioning (Fig. 1A).

### Plant materials and light treatment

Two lettuces (*Lactuca sativa* L. cv. 'Spanish green' and 'Butterhead') were cultivated using hydroponic methods. The seeds were germinated in a seedling room for 2 days (no light, temperature 18 ± 2 °C, humidity: 70 ± 5%), and then transferred to foam boards to initiate supplementary lighting., then transplanted into the foam boards and supplementary lighting begins. The experiment included eight distinct light intensities, ranging from 125 to 300 µmol/m$^2$/s, labeled as P125 (125 µmol/m$^2$/s), P150 (150 µmol/m$^2$/s), P175 (175 µmol/m$^2$/s), P200 (200 µmol/m$^2$/s), P225 (225 µmol/m$^2$/s), P250 (250 µmol/m$^2$/s), P275 (275 µmol/m$^2$/s), and P300 (300 µmol/m$^2$/s). Each group was grown under the corresponding light intensity for 38 days before harvesting (Fig. 1B). During cultivation period, the laboratory temperature was maintained at 23 ± 2 °C, and the selected nutrient solution was a commercial formula from Otsuka in Japan, with an electrical conductivity (EC) value of 2.6 mS/cm.

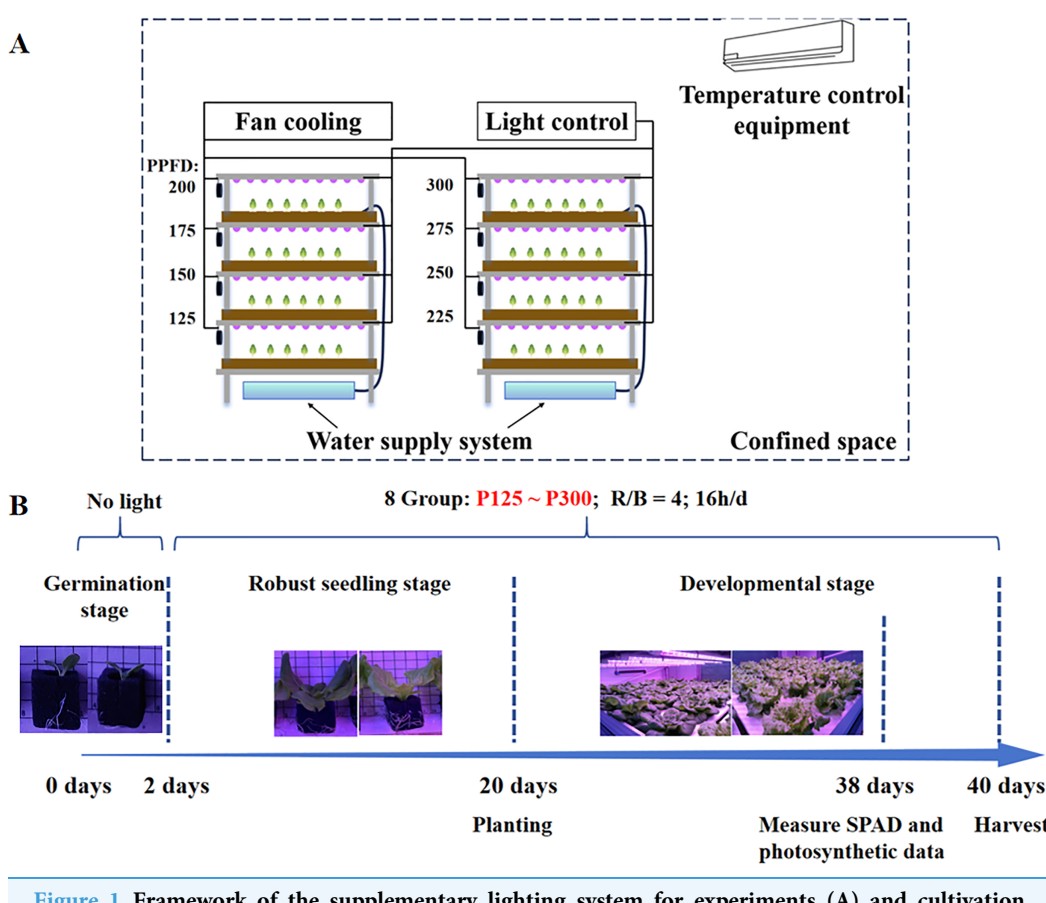

**Figure 1 Framework of the supplementary lighting system for experiments (A) and cultivation process (B).**

Each layer of supplemental lighting fixtures comprises 12 lamp tubes of the T8 type, with a fixed spacing of 20 cm between each tube. The lamp tubes contain red (660 nm) and blue (450 nm) LED chips, with the chip packaging type being 2,835. The red and blue LED channels of the lamp can be independently adjusted. In this experiment, the power output ratio of red to blue light was set at 4:1 (R4B1), with a light/dark cycle of 16/8 h (0:00–16:00).

## Measurement of uniformity of illumination

To measure the distribution of light intensity from the supplemental lighting fixtures, a device for testing light uniformity and an upper control system were designed. The light uniformity testing device consists of a spectrometer and a mobile cart, with a total volume of 20 cm × 14 cm × 20 cm. The measured light intensity data from the spectrometer is transmitted to the host computer system *via* Wi-Fi. Automatic measurements are conducted at a height of 30 cm below the lighting fixtures. A total of 63 measurement points (nine points in length and seven points in width) are collected for the experiment (Fig. 2). The measuring cart employs PID control and a gyroscope for calibration, ensuring the accuracy of the measurement route. The number of measurement points is controlled by the input from the host computer system. After the measurement is complete, a PPFD

![PeerJ]

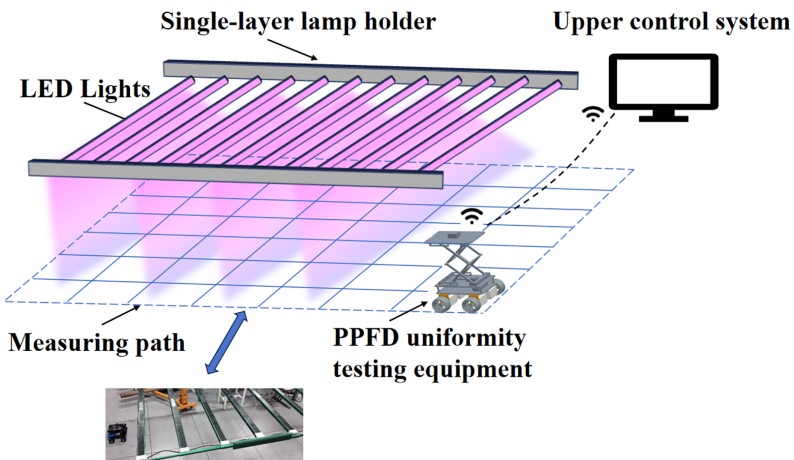

**Figure 2** Schematic diagram of PPFD uniformity detection and measurement results.

distribution map and uniformity index are generated. Based on the value of uniformity, the angle of each layer of T8 lamp tubes can be adjusted to improve uniformity. Use the measuring vehicle to conduct measurements and repeat this process until all groups achieve a uniformity of at least 80%.

## Control method for detection equipment

The framework of the measurement device is shown in Fig. 3. Communication with the user's end is established through Bluetooth, and the data measured by the spectrometer is transmitted to the client *via* Wi-Fi. The control center utilizes the STM32F103 series chip as its core. The operational speed of the device is controlled through a PID (Proportion Integral Differential) algorithm, ensuring precise speed control. Additionally, the MPU6050 gyroscope is employed for correction to prevent inaccuracies in the measurement route; After adjusting these two parameters, PWM waves are generated to control the operation of the equipment. Control commands are transmitted *via* Bluetooth, and the equipment plans measurement points based on the input measurement information. The measurement route follows an "S" shape to facilitate the plotting of result data. Measurement results are displayed on the user end as a map of PPFD uniformity distribution along with the uniformity value. Researchers adjust the corresponding areas of T8 lamps based on the distribution map. An OLED display module is used to show current measurement point information, and an independent power source supplies power to the entire measurement device.

## Measurement and data analysis

Samples were taken on days 20, 25, 30, 35, and 40. Four plants were randomly selected from each experimental group. The leaf length, leaf width, and leaf amplitude of the second fully expanded true leaf were measured using a ruler. Leaf amplitude data refers to the maximum length of the vertical projection of the plant. Measurements were concluded on day 35.
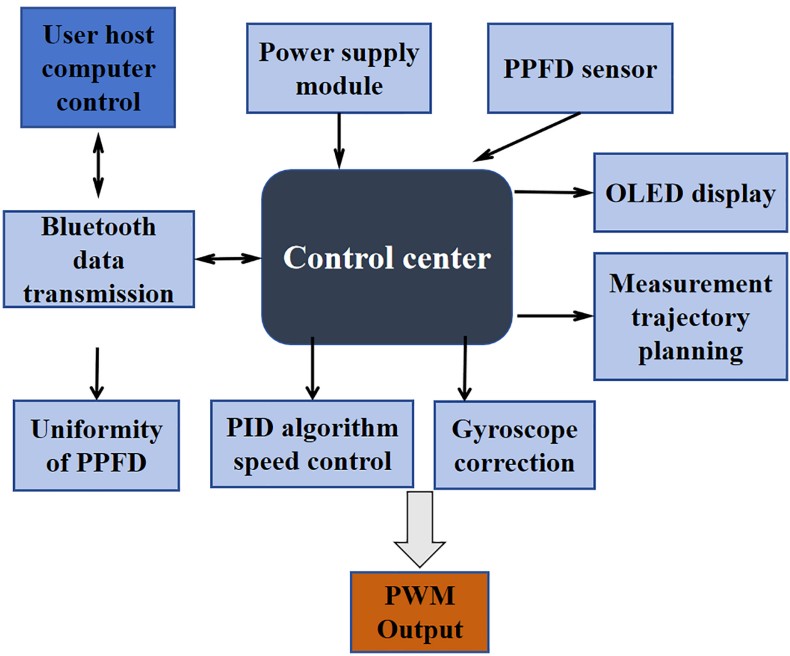

**Figure 3 Framework of the automatic detection device and human-machine interaction process.**

On the 38th day after sowing, the photosynthetic rate (Pn), transpiration rate (Tr), and stomatal conductance (Gs) of the second fully expanded true leaf were measured using a portable photosynthesis meter (LI-6400; Li-COR, Lincoln, NE, USA). Four plants from each group were randomly selected for measurement. Simultaneously, the SPAD value of the second leaf was measured using a chlorophyll meter (SPAD-502PLUS; Konica Minolta, Tokyo, Japan), with three repetitions of the measurement. Three different positions are selected to measure a fully expanded leaf. After harvest (40 d), the fresh weight (FW) of all lettuce plants was measured using an electronic balance (JX-C, JINXUAN, CN). Then, four lettuce plants from each group were selected, placed in an oven at 105 °C for blanching for 1 h, and the oven temperature was adjusted to 60 °C until a constant weight was reached. The dry weight (DW) of the lettuce is then measured. The dry matter content (DMC) is calculated as the ratio of the dry weight to the fresh weight. A power meter was installed for each layer of light fixtures to measure the actual power consumption of each layer.

$$\mathrm{PER}_i = \frac{W_{iavg} \times \dfrac{W_i}{W_{max}}}{P_i} \tag{1}$$

The production efficiency ratio (PER) is expressed as the ratio of the weighted average weight per group to the total power consumption of each layer of lights. Here is the specific formula, where $W_{iavg}$ represents the average weight of each group, $W_i$ represents the weight of three randomly selected plants in each group, $W_{max}$ represents the maximum weight among $W_i$ across all groups, and $P_i$ represents the power consumption of each group.

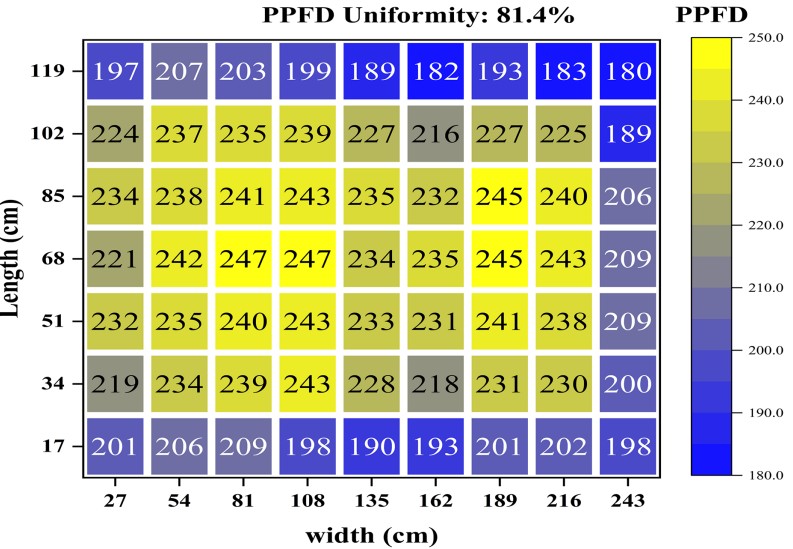

**Figure 4  PPFD distribution of the supplementary lighting plane in the P225.**

The data is processed using SPSS 22 software for analysis of variance (ANOVA), followed by Duncan's multiple range test at a 95% confidence level ($\alpha = 0.05$) to assess significant differences.

## RESULT

### PPFD distribution

In terms of the lighting uniformity of the luminaires in the eight experimental groups, measurements were conducted using automatic detection equipment and an upper computer. The results showed that the uniformity was above 80% before proceeding with the experiments. Taking Fig. 4 as an example, it reflects a PPFD uniformity of 81.4% at P225. There are significant PPFD variations at the boundaries, with the minimum value being 180 μmol/m$^2$/s. When producing and cultivating crops in plant factories, the supplementary lighting plane tends to have lower light intensity at the edges and excessively high light intensity in certain inner areas, leading to inconsistent growth. To avoid this situation, the experimental lettuce was planted in the inner area.

### The growth morphology of lettuce

In the leaf amplitude data of both lettuce varieties, it is evident that the rate of increase in leaf amplitude is highest between days 20 and 25. In Butterhead lettuce, P250 exhibits the lowest leaf amplitude on the 20th day, while P125 shows the highest leaf amplitude. However, on day 35, P225 reaches its maximum leaf amplitude (Fig. 5A). For Spanish green lettuce, under the P250 condition, it consistently exhibits the smallest leaf amplitude, while under the P125 condition, the leaf amplitude is greater than that of the other experimental groups (Fig. 5B). This lettuce variety shows distinct responses of contraction and expansion to light intensity.

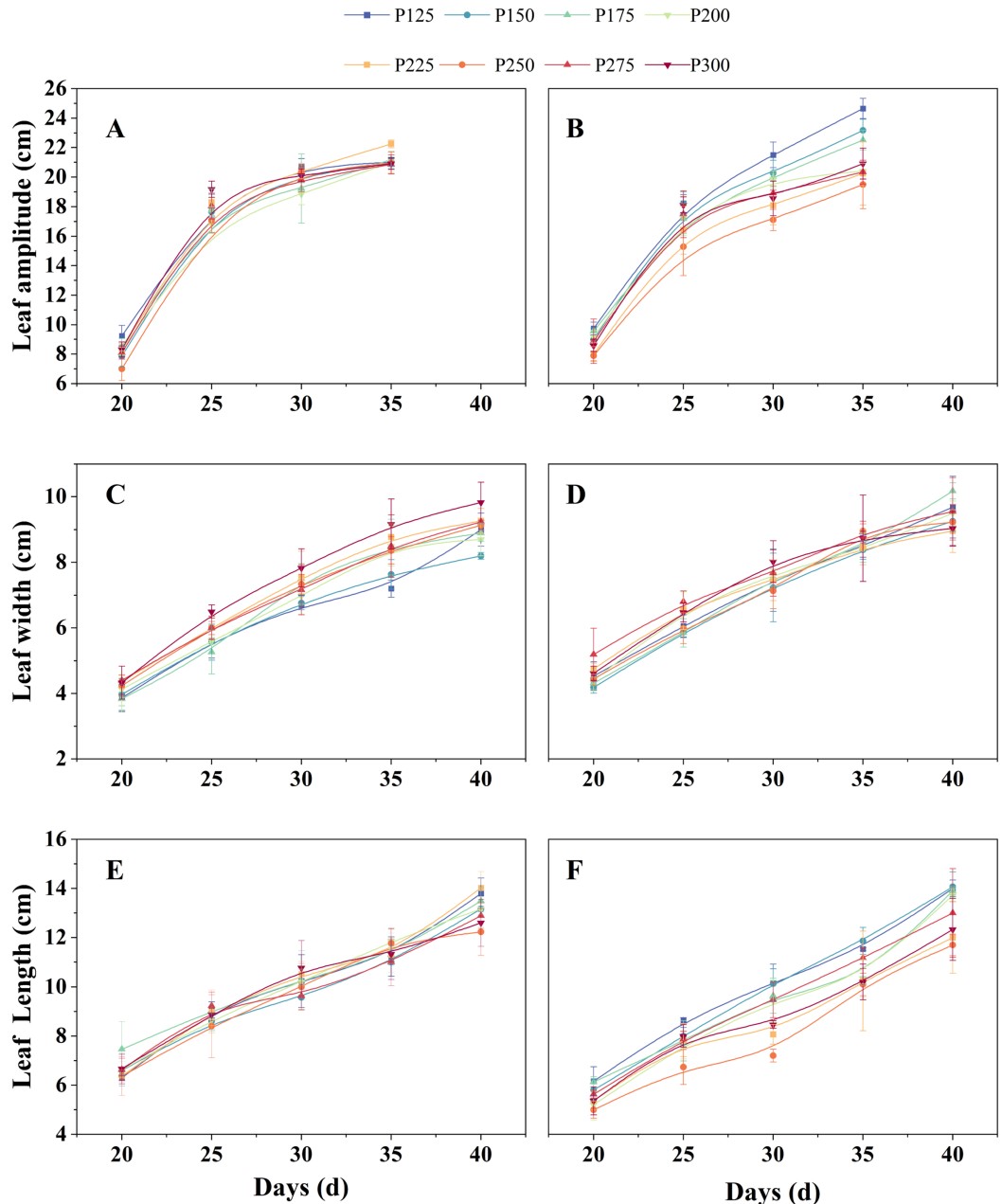

**Figure 5 The influence of different light intensities on the leaf amplitude, leaf width, and leaf length of butterhead lettuce (A, C, E) and Spanish green lettuce (B, D, F) under the same light quality ratio.**

In terms of leaf length and width, variations in light intensity significantly affect the leaf width of Butterhead lettuce (Figs. 5C, 5E). Under the P300 condition, it consistently maintains the widest leaf width. On the 40th day, P225 exhibits the longest leaf length. During the growth process of Spanish green lettuce, variations in light intensity have a significant impact on its leaf length (Fig. 5F). Under the P125 and P150 light intensity conditions, the leaf length consistently remains the longest. However, under these two conditions, the leaves are thin and weak. On the 40th day, under the P200 condition, the

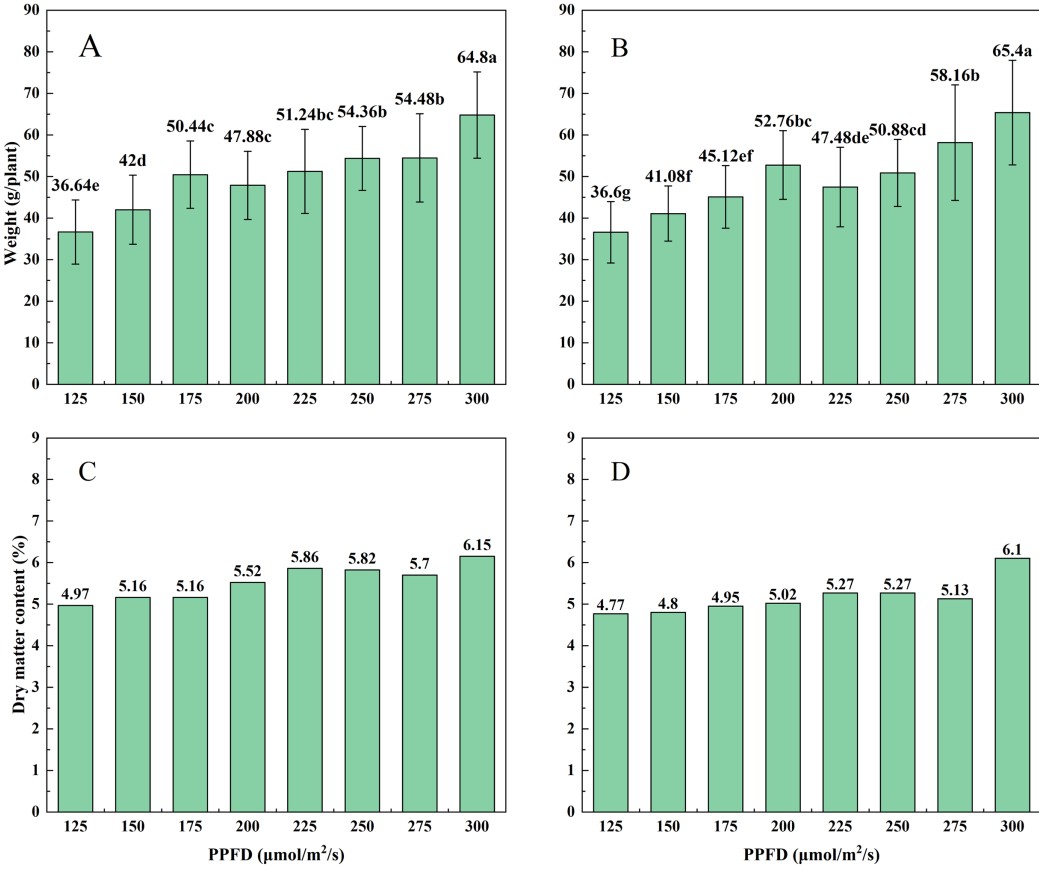

**Figure 6 The influence of different light intensities on the weight and dry matter content of butterhead lettuce (A, C) and Spanish green lettuce (B, D) under the same light quality ratio.**

leaf length of Spanish green lettuce is only surpassed by P125 and P150, while the leaf width is greater than that of the other groups (Fig. 5D). Additionally, for both lettuce varieties, under the P125, P150, and P175 conditions, they exhibit weak, thin plant states, which are unfavorable for lettuce production.

## Fresh weight, dry matter content, and production capacity ratio

After 40 days of cultivation, both varieties of lettuce exhibit the same trend, with fresh weight increasing with increasing light intensity. At a light intensity of 125 $\mu mol/m^2/s$, the fresh weights of Butterhead lettuce and Spanish green lettuce reach their lowest values, at 36.64 and 36.60 g. At a light intensity of 300 $\mu mol/m^2/s$, both Butterhead lettuce and Spanish green lettuce reach their maximum fresh weights, at 64.80 and 65.40 g respectively (Figs. 6A, 6B). Furthermore, for Spanish green lettuce, at a light intensity of 200 $\mu mol/m^2/s$, the fresh weight is 52.76 g, significantly higher than the P175 and P225 experimental groups. Regarding the accumulation of dry matter, both Butterhead lettuce and Spanish green lettuce reach their maximum at 300 $\mu mol/m^2/s$, at 6.15% and 6.10% respectively (Figs. 6C, 6D). This indicates that at a light intensity of 300 $\mu mol/m^2/s$, the influence on the accumulation of dry matter is greater for both lettuce varieties.

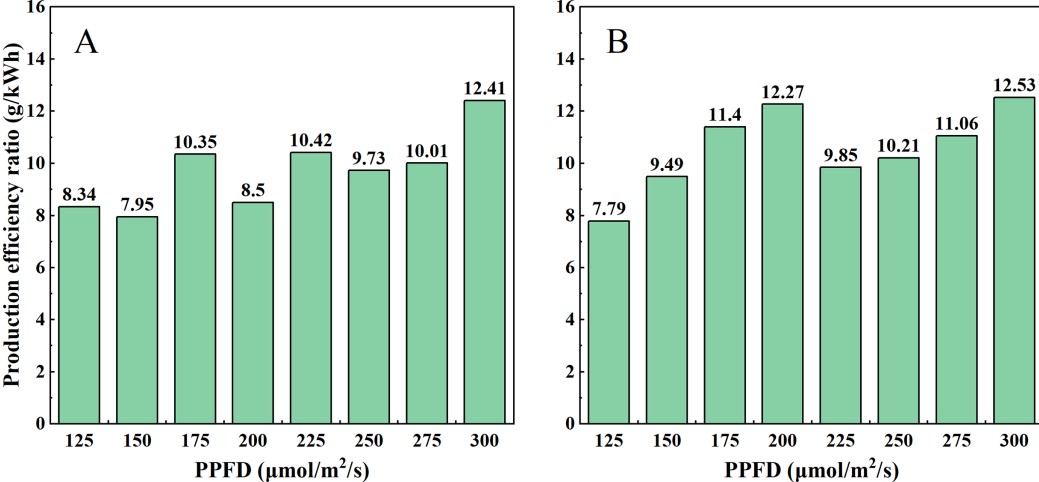

**Figure 7** Changes in production efficiency ratio of (A) Butterhead lettuce and (B) Spanish green lettuce.

After harvesting the two lettuce varieties, the production efficiency ratio of different experimental groups is calculated to facilitate determining the suitable light intensity for lettuce production. For Butterhead lettuce, under the P300 experimental condition, the production efficiency ratio reaches its maximum at 12.41 g/kwh, which is advantageous for Butterhead lettuce production (Fig. 7A). For Spanish green lettuce, under the P300 and P200 experimental conditions, the production efficiency ratios are 12.53 and 12.27 g/kwh respectively, with the former being 2.12% higher than the latter. This provides reference significance for lettuce production of Spanish green lettuce (Fig. 7B). In terms of electricity consumption, the daily electricity consumption of P300 and P200 is 5.22 kWh and 3.6 kWh, respectively, P300 consumes 45% more energy per day than P200. Therefore, P200 conditions can be used to cultivate Spanish green lettuce.

## SPAD and leaf photosynthetic

The SPAD value reflects the relative chlorophyll content of plants and serves as a reference for assessing the chlorophyll content of plants. For Butterhead lettuce, the overall trend of SPAD value increases with increasing light intensity (Fig. 8A). For Spanish green lettuce, the SPAD value reaches its maximum at P225 (Fig. 8B). This indicates that under P225 conditions, it is beneficial for the increase of chlorophyll in Spanish green lettuce

Figure 9 depicts the photosynthetic performance of the two lettuce varieties under different light intensities. Under the P300 condition, the Butterhead lettuce and Spanish green lettuce exhibit mean Pn values of 9.90 and 10.64 $\mu mol/m^2/s$ respectively, However, under the P125 condition, the values are 7.49 and 8.86 $\mu mol/m^2/s$ respectively (Figs. 9A, 9B). This indicates that the photosynthetic rate of both lettuce varieties is higher under the P300 condition. In terms of Tr data, Butterhead lettuce exhibits discrete values under the P225 and P275 conditions (Fig. 9C), while Spanish green lettuce shows no discrete values. Under the P275 and P300 conditions, the interquartile range (IQR) for Spanish green lettuce is 2.19 and 2.32 $mmol/m^2/s$ respectively, with mean values of 20.31 and

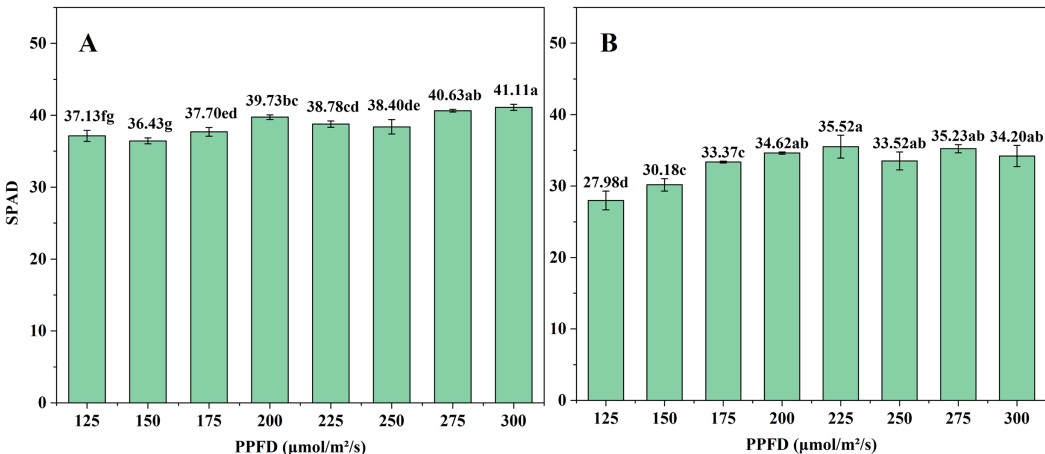

**Figure 8** The effect of different light intensities on the SPAD values of (A) Butterhead lettuce and (B) Spanish green lettuce.

20.61 mmol/m$^2$/s respectively. This indicates that the transpiration rate of Spanish green lettuce is generally higher under the P300 condition (Fig. 9D). In terms of Gs data, the median values for both lettuce varieties under the P300 condition are 0.36 and 0.34 mmol/m$^2$/s respectively, indicating that within the range of light intensities studied, P300 exhibits the highest stomatal conductance (Figs. 9E, 9F).

## DISCUSSION

In most studies on supplemental lighting for lettuce, the main focus is on the effects of light quality, intensity, and photoperiod. However, in actual production, it is also important to consider light uniformity, as this ensures that the overall data is more representative. In this study, an automatic PPFD uniformity measurement device was designed to ensure that the uniformity of each experimental group is above 80% before conducting lettuce light intensity experiments. In terms of leaf amplitude growth, both varieties of lettuce experienced rapid increases between days 20 and 25. This indicates an increased demand for light during this period, and the increase in leaf amplitude is beneficial for plants to capture more light. Additionally, the increase in leaf amplitude may also be a response to the crop being under low light stress, necessitating the enlargement of leaf amplitude to capture more light energy to sustain growth requirements. In general, plants tend to be robust in high light intensity environments and thin in low light intensity environments (*Fu et al., 2017*). In the study, it was observed that when the supplemental lighting intensity was below 175 µmol/m$^2$/s (P175, P150, P125), Butterhead lettuce and Spanish green lettuce exhibited poor growth, narrow leaf width, and elongated leaf length, indicating that growth was hindered under this light intensity.

In the physiological indices of lettuce, weight is the core evaluation criterion. The results of this study indicate that within the light intensity range of 125 to 300 µmol/m$^2$/s, the fresh weight of Butterhead lettuce increases with increasing light intensity. In the research of *Carotti et al. (2021)* on lettuce, it was also observed that with increasing light intensity (200, 400, and 750 µmol/m$^2$/s), the fresh weight also increases. However, the situation

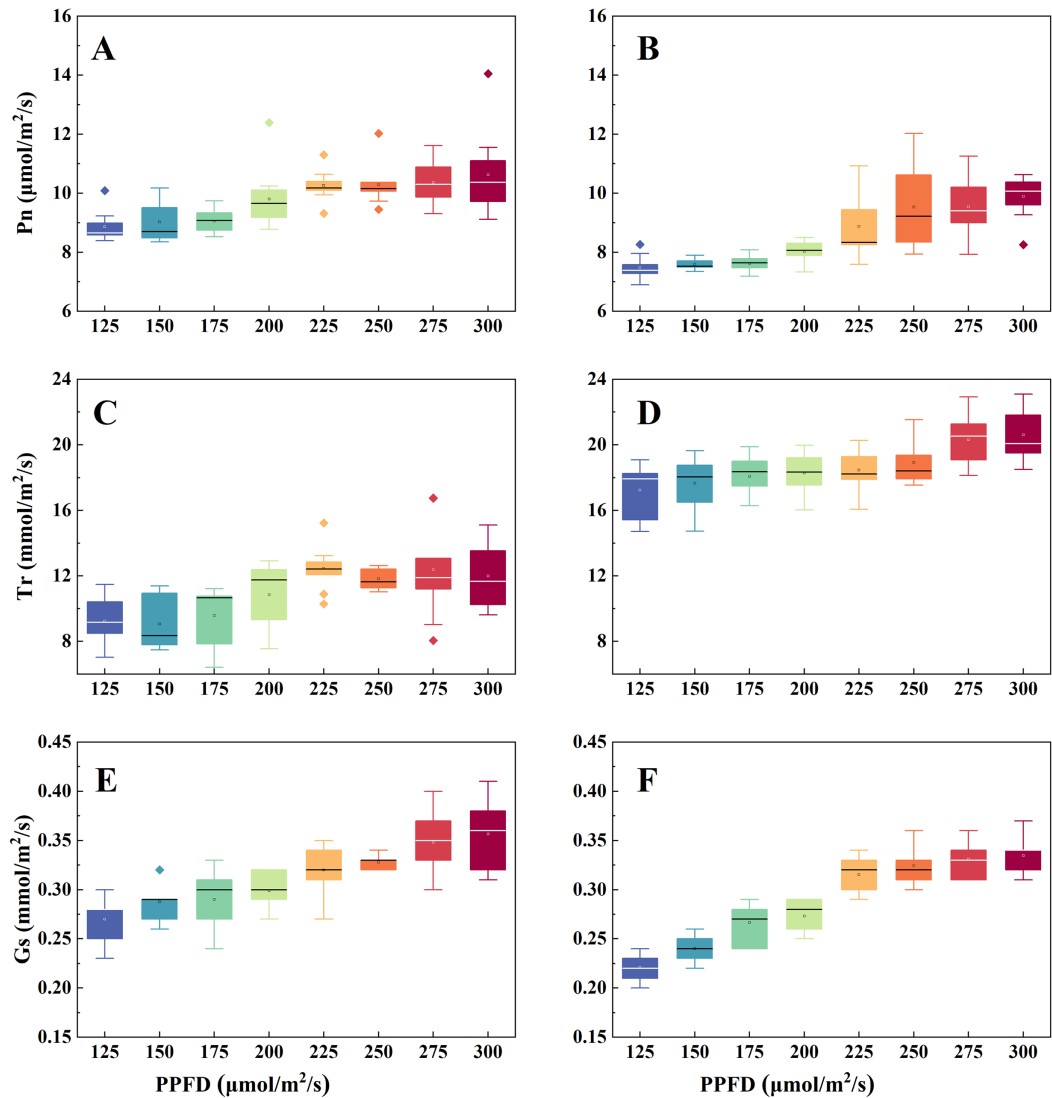

**Figure 9 Photosynthetic performance of butterhead lettuce (A, C, E) and Spanish green lettuce (B, D, F) under different light intensities.**

beyond 300 μmol/m²/s requires further investigation. Its dry matter content also shows a slow upward trend with increasing light intensity. According to research by *Larsen et al. (2020)*, an increase in light intensity is associated with an increase in the dry matter content of basil. In terms of fresh weight, Spanish green lettuce shows a significant difference in experimental group P200 compared to groups P175 and P225. The appearance of a minor peak at this point suggests that 200 μmol/m²/s may be a favorable value for production. It can be observed that Butterhead lettuce exhibits a higher production efficiency ratio under a light intensity of 300 μmol/m²/s. This indicates that under equal electricity consumption, better quality can be produced. Spanish green lettuce shows similar results at 200 and 300 μmol/m²/s, indicating that it may be preferable to conduct daily production at 200 μmol/m²/s, which is advantageous for cost reduction. In *Fu, Li & Wu (2012)*, it is

mentioned that although the highest fresh weight occurs at 600 µmol/m$^2$/s, the highest light utilization efficiency is observed at 200 µmol/m$^2$/s.

Chlorophyll primarily absorbs red and blue light wavelengths (*Johkan et al., 2010*). The SPAD value of leaves is used as a useful indicator for assessing the chlorophyll content of leaves (*Mendoza-Tafolla et al., 2019*; *Dong et al., 2019*; *Donnelly et al., 2020*). For ease of measurement, this study utilizes relative chlorophyll to assess the differences between each experimental group. Figure 8 shows that SPAD values are lower when below 200 µmol/m$^2$/s, indicating lower chlorophyll content in lettuce leaves under low light intensity. This indicates a reduction in photosynthesis, leading to decreased yield. Results from *Zhou, Li & Wang (2022)* indicated that the content of chlorophyll also influences the final yield to some extent. With increasing light intensity, both the chlorophyll content and the yield also increase.

Low light intensity may reduce the production of ATP and NADPH, decrease the activity of carbon assimilation enzymes, and reduce the number of chloroplasts, thereby restricting plant carbon assimilation and photosynthetic rate (*Zheng et al., 2011*; *Pires et al., 2011*; *Fu, Li & Wu, 2012*; *Da Silva Branco et al., 2017*). Therefore, this often leads to low yield and reduced photosynthetic rates in most plants. In treatments P125, P150, and P175, both lettuce varieties exhibited lower Pn compared to the other groups, which is also reflected in the final yield. With the light intensity increases, Gs values also increase, reflecting that within a certain range of light intensity, increasing light intensity benefits stomatal development. The photosynthesis of leaves is regulated by stomata, and stomatal opening is a general response to high light intensity (*Mancarella et al., 2016*). Previous studies have indicated that when the PPFD increases from 60 to 220 µmol/m$^2$/s, the stomatal conductance of lettuce also increases accordingly (*Fu et al., 2017*). In addition, some studies have also pointed out that an increase in light intensity is beneficial for the development of lettuce stomata (*Ghorbanzadeh et al., 2021*). At the same time, this also results in an acceleration of transpiration rate (Tr). The study suggests that the increase in light intensity promotes photosynthesis, leading plants to require more photosynthetic materials. This results in an increase in Gs and Tr values. Overall, light intensity significantly influences the photosynthesis (Pn), transpiration rate (Tr), and stomatal conductance (Gs) of both lettuce varieties, with all showing an increasing trend with increasing light intensity. For Butterhead lettuce, the mean values of Pn and Gs reach their maximum under the P300 experimental condition, and Spanish green lettuce exhibits similar characteristics. This is consistent with the final harvest weight, where increased light intensity promotes photosynthesis, leading to accumulation of fresh weight and dry matter.

## CONCLUSION

In supplemental lighting for vertical farm, it's essential to consider the uniformity of the lighting fixtures, which contributes to overall efficient production. This study designed an automated device to test the uniformity of lighting fixtures, ensuring that the uniformity of supplemental lighting is above 80%, thus making the growth data of lettuce more representative. The study investigated the effects of different light intensities on the

physiology, photosynthesis, and productivity of two lettuce varieties, with a red-blue ratio of 4:1 (R4B1). The study indicates that both lettuce varieties experienced a significant increase in leaf amplitude between the 20th and 25th days of growth, which is a favorable period for altering crop morphology. Additionally, the study concludes that under a condition of 300 μmol/m$^2$/s, Butterhead lettuce exhibits optimal fresh weight and production efficiency ratio. Spanish green lettuce exhibits a production efficiency ratio of 12.53 g/kwh under 300 μmol/m$^2$/s conditions, while under 200 μmol/m$^2$/s conditions, the production efficiency ratio is 12.27 g/kwh, showing a difference of only 2.12% between the two light conditions. This study suggests that in plant factories, Spanish green lettuce should be cultivated under a light intensity of 200 μmol/m$^2$/s when the red-to-blue light ratio is R/B = 4, while Butterhead lettuce can be supplemented with a light intensity of 300 μmol/m$^2$/s.

### Funding
This work was supported by the Shanghai Science and Technology Committee (STCSM) Science and Technology Innovation Program (Grant Nos. 23N21900100, 22N21900400). The funders had no role in study design, data collection and analysis, decision to publish, or preparation of the manuscript.

### Grant Disclosures
The following grant information was disclosed by the authors:
Shanghai Science and Technology Committee (STCSM): 23N21900100, 22N21900400.

### Competing Interests
The authors declare that they have no competing interests. Yan Shen is employed by Shanghai Yingzhi Technology Co., Ltd.

### Author Contributions
- Jun Zou conceived and designed the experiments, authored or reviewed drafts of the article, and approved the final draft.
- Shipeng Luo conceived and designed the experiments, performed the experiments, analyzed the data, prepared figures and/or tables, and approved the final draft.
- Mingming Shi analyzed the data, authored or reviewed drafts of the article, and approved the final draft.
- Dawei Wang performed the experiments, analyzed the data, prepared figures and/or tables, and approved the final draft.
- Wenbin Liu analyzed the data, authored or reviewed drafts of the article, and approved the final draft.
- Yan Shen performed the experiments, prepared figures and/or tables, and approved the final draft.
- Xiaotao Ding conceived and designed the experiments, prepared figures and/or tables, authored or reviewed drafts of the article, and approved the final draft.

- Yuping Jiang conceived and designed the experiments, authored or reviewed drafts of the article, and approved the final draft.

## Data Availability

The raw measurements are available in the Supplemental File.

## Supplemental Information

Supplemental information for this article can be found online at http://dx.doi.org/10.7717/peerj.19229#supplemental-information.

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
