# Peer review of "Effects of different light intensities on lettuce growth, yield, and energy consumption optimization under uniform lighting conditions"

_PeerJ, doi:10.7717/peerj.19229_

## Round 0.1 · original submission · Major Revisions

Vertical farming is a promising area of study with immense potential for cultivating vegetables that grow rapidly and are particularly susceptible to pests and diseases. Among the critical factors in this approach, light plays a critical role. Your study provides significant knowledge to the lettuce grower seeking to optimize energy consumption. However, it is essential to address certain technical details to enhance the article further. I encourage you to carefully review the reviewers' suggestions and thoughtfully consider each recommendation. If you find yourself in disagreement with any specific suggestions, providing a clear and well-supported rationale for your viewpoint would be highly beneficial.

·

Basic reporting

There are areas where language and grammar can be improved. It is recommended that the authors rephrase the following sentences for clarity:

• Line 38 – 41: “Hydroponic is one of the commonly …. provide a strong basic guarantee for the development of facility agriculture”.
• Line 55: “… inter-plant lighting in the closed artificial light”.
• Line 57:”… in terms of shape, the lamps have bar, round and square and the representative of the bar lamps is the long straight tube lamp, …”
• Line 72: “… study by Maria Grzegorzewska. and showed … “

Line 49: “Spectrum adjustment technology can help to reduce product costs …” should be backed by a citation.

Figure 5: The x-axes labels should be Days instead of data.

Experimental design

The authors applied some form uniformity correction to the light source to ensure good lighting uniformity in the setup and use this setup to optimize the PAR intensity required to grow two variants of lettuce in a vertical cultivation system.

It will be good for the authors to clarify on some of the experimental details listed below:

• In this study, each tier is illuminated by a set of “single-layer lamp holder”. Can the authors share more details on how the irradiation angles of the LED tubes are adjusted? Are the individual LED tubes tilted independently, or the entire lamp holder is tilted to improve uniformity?

• Figure 2 shows that the optimization of the lighting uniformity is done on the floor and not directly at the vertical farming racks. What are the measures that the authors take to ensure that the optimized settings are transferred to the respective layers?

• The authors should share in more details how the plants are selected for measurement. Is the sampling of the crops random or predetermined? Are the chosen plants’ position consistent across all the layers?

• Model and brand of the equipment and tools used in this study should be described in more details to provide readers sufficient information to replicate the experiments. For example, model and brand for photosynthesis meter and SPAD meters are missing.

Validity of the findings

In my opinion, this study might be more impactful if there are lighting uniformity data of the LEDs before the adjustments of angles (baseline configuration). In addition, the corresponding crop yield/production efficiency ratio data of this baseline configuration should also be provided. The authors may consider adding these data to justify the advantage of using their tool/protocol to improve the uniformity of the luminaire.

Additional comments

No comments

·

Basic reporting

no comment about this section.

Experimental design

I have identified some design errors in the test environment prepared for vertical farming. In this study, the effects of light intensity on the growth and yield of test plants (two different types of lettuce) were investigated experimentally. Light wavelength affects the growth and yield parameters of plants as much as light intensity. Therefore, the wavelength of the light applied to the plant should also be taken into account in this application. At this point, the study can be transformed into two ways. The first way is wavelength stability. Additional control measures should be taken to prevent the wavelength from changing as the light intensity increases. For this, flexible red and blue LED groups should be created and should be activated and deactivated in a control loop. The second way is to examine the effects of wavelength and light intensity effects on plant growth and yield, which is also my recommendation. Otherwise, I believe that correct analyses cannot be made with the data obtained from the application system of the study.

Validity of the findings

It was observed that the number of light sources determined for the PPFD unifomity analysis (12 units) and the partitioned test area dimensions were designed between 2.6 m (length) × 1.28 m (width). It is understood from Figure 4 that the data obtained (7x9 partitions) were taken. A proportional imbalance was detected in the data obtained with this method. With this partitioning, it is difficult to apply the same light to 84 root plants and I foresee that it will negatively affect the accuracy of the light analysis. Instead, I recommend that it be revised with an arrangement and measurement system that will guarantee that each light tube is placed vertically below.

·

Basic reporting

This article explores the potential applications of artificial light in vertical farming within plant factories, a topic of considerable interest. A comprehensive revision of the English language within the manuscript is strongly recommended.

Experimental design

The experimental design is suitable for this journal, but the description of the experimental method requires more details. This study did not specify the specific experimental design in the main text.

Validity of the findings

The article lacks detailed discussion of energy consumption. The author should elaborate on energy use during cultivation and compare it with conventional greenhouse farming without artificial light.
Since the authors want to express the optimization of energy consumption, please discuss more about how much the cost of energy reduction varies according to the different light intensity.

1. Please carefully check the format and order of the references.
2. Please check for all technical terms and abbreviations that are clearly defined.
3. Please check the graphs that should be set as the same scale and values where comparisons are intended because uniform scales across graphs will improve readability and make comparisons between data points clearer and more meaningful.
4. Consistency in the units of measurement throughout the manuscript should be ensured. For example: PPFD unit, some are in µmol/m2/s and some are in µmol m-2.s-1
5. Please add the scientific name of the lettuce and the exact cultivar or variety name. What is “Spanish green lettuce”?
6. Chlorophyll data for each observation should be presented, as chlorophyll content varies across the crop's growth stages.
7. If possible, the author could calculate the growth rate. The growth rate equation is provided in the following article as a reference: [doi:10.4081/jae.2024.1579].
8. Please check for “Leaf amplitude”. Should it be lower case or not?
9. The author should consider a comparison with other studies on the use of artificial light in lettuce cultivation. Factors such as the red/blue light ratio and dark/day cycles significantly influence outcomes. Presenting a summary in table form is highly recommended.
10. Figure 4: Please add an explanation about what P225 is and add the unit of PPFD. How many number of samples?
11. Please provide a better solution for Figure 5 and add an explanation about the error bars. What is the data on the x-axis? Is it days after sowing? How many number of samples?
12. Figure 6: Please add values, and the unit is g/plant or not (on the y-axis). Also, please give an explanation about the error bars and different alphabets. How many number of samples? It would be better if the authors can provide the dry weight.
13. Figure 7: Should the unit be the same style?
14. Figure 8: Please add values and explain the different alphabet.
15. Line 113: The nutrient requirements of plants vary at different growth stages. Line [113] states that the nutrient solution for lettuce cultivation has an EC of 2.6 mS/cm. The author should clarify whether different nutrient solution concentrations were provided during the stages of plant growth or if a constant concentration was used throughout the cultivation process.
16. Line 130-131: Before conducting the supplemental lighting experiment, the light uniformity of each experimental group is adjusted to above 80%. Please check this sentence.
17. Line 132: please explain about PID control.
18. Line 158: On the 38th day after sowing, photosynthetic parameters of the plants are measured using a photosynthesis meter, with measurements conducted under each group's respective light intensity conditions. Which measurements and which values will they be?
19. Line 159: S Additionally, SPAD meters are used to record the SPAD values of each group of lettuce. Is it a SPAD-502 Chlorophyll meter?
20. Line 162-163: Subsequently, four lettuce plants from each group are selected and placed in an oven at 105°C until a constant weight is achieved. I think the temperature should be 65°C or not?
21. Line 168-170: These sentences can be placed after the equation, and the authors can add the equation number.
22. Line 172-173: The sentences are not completed. Please check!
23. Line 194-202: Please check these sentences. The word “slender” seems to be not correct word.
24. Line 203: Fresh weight, Dry Matter Content, and Production Capacity Ratio. Please keep the format the same, whether it is upper case or lower case.
25. Line 209-210: Please check the sentence. It would be good if data of dry weight can be added. The dry matter content values are average values or not?
26. Line 212: at 6.15% and 6.1%, respectively (Fig.6C, D). Please maintain consistency in the decimal points.
27. Line 205-212: The fresh weight values are in the unit of g. Are they the average values of the samples per treatment?
28. Line 221: The difference of 2.11% between the two. Please check the values!
29. Line 223: No methodology on Leaf Photosynthetic Net Rate calculation mentioned before.
30. Line 224-227: The author states that increased light intensity enhances chlorophyll measurements on SPAD; however, Figure 8 does not reflect this, showing more fluctuating SPAD values. The author should clarify this. The increasing trend is observed in butterhead but not in Spanish green lettuce.
31. Line 228-240: No explanation in the methodology, and some abbreviations appear without being mentioned before. A table with the values should be provided so that the reader will easily understand. Using the average to compare different treatments would be better than using the median. In Figure 9, the explanation about the boxplot must be prepared, such as median line, whiskers, dots (outliers), and so on. The x-axis is the PPFD?
32. Line 241-246: These lines should be in the discussion.
33. Line 262: However, Spanish green lettuce did not exhibit this phenomenon. Is it at the same light intensity?
34. Line 263-264: Is the format of citation correct? The study does not contain any data on the growth rate.
35. Line 267-269: Is the format of citation correct?
36. Line 271-272: Is the citation format correct?
37. Line 280-282: Is the citation format correct?
38. Line 289-291: Is the citation format correct?
39. Line 303-304: In addition, some studies have also pointed out that an increase in light intensity is beneficial for the development of lettuce stomata. Please cite these!
40. Line 314: The study investigated the effects of different light intensities on the physiology, photosynthesis, and productivity of two lettuce varieties, with a red-blue ratio of 8:2. This “red-blue ratio of 8:2” did not mention in the methodology.

---

## Round 0.2 · accepted · Accept

I would like to thank you for accepting the referees' suggestions and improving your article based on their suggestions. Your article is ready to publish. We look forward to your next article.

The Section Editor noted:

> In the Abstract, manuscript text, and Figure 5 the authors have used a cryptic term "leaf amplitude" which is not used in botany or biological studies. The authors seem to be referring to plant height. A plant's height is not an amplitude - I think this is a language translation problem. Why not simply refer to plant height throughout the manuscript?

·

Basic reporting

Literature references and field background/context have been sufficiently revised.
The paper was supported by results consistent with the hypotheses as a result of revisions.

Experimental design

In the experimental application part of the paper, invalid value and setup details in the light intensity distribution/measurement systematics were revised.

Validity of the findings

Comparative analysis of the results obtained in the study reveals the study in its true sense. Based on this, the revisions made were analyzed with product efficiency ratio and the effects of light intensity on the product growth process and its contribution to efficiency.

Additional comments

As a result of the suggested revisions, I think the article has achieved its intended conclusion.